# UIFace: Unleashing Inherent Model Capability to Enhance Intra-Class Diversity in Synthetic Face Recognition

**Xiao Lin**[1,*]**, Yuge Huang**[1,*]**, Jianqing Xu**[1]**, Yuxi Mi**[2]**, Shuigeng Zhou**[2]**, Shouhong Ding**[1,†]
[1]Tencent Youtu Lab    [2]Fudan University
`{xiaolin,yugehuang,joejqxu,ericshding}@tencent.com`
`{yxmi20,sgzhou}@fudan.edu.cn`

## ABSTRACT

Face recognition (FR) stands as one of the most crucial applications in computer vision. The accuracy of FR models has significantly improved in recent years due to the availability of large-scale human face datasets. However, directly using these datasets can inevitably lead to privacy and legal problems. Generating synthetic data to train FR models is a feasible solution to circumvent these issues. While existing synthetic-based face recognition methods have made significant progress in generating identity-preserving images, they are severely plagued by context overfitting, resulting in a lack of intra-class diversity of generated images and poor face recognition performance. In this paper, we propose a framework to **U**nleash **I**nherent capability of the model to enhance intra-class diversity for synthetic face recognition, shortened as **UIFace**. Our framework first trains a diffusion model that can perform sampling conditioned on either identity contexts or a learnable empty context. The former generates identity-preserving images but lacks variations, while the latter exploits the model's intrinsic ability to synthesize intra-class-diversified images but with random identities. Then we adopt a novel two-stage sampling strategy during inference to fully leverage the strengths of both types of contexts, resulting in images that are diverse as well as identity-preserving. Moreover, an attention injection module is introduced to further augment the intra-class variations by utilizing attention maps from the empty context to guide the sampling process in ID-conditioned generation. Experiments show that our method significantly surpasses previous approaches with even less training data and half the size of synthetic dataset. The proposed **UIFace** even achieves comparable performance with FR models trained on real datasets when we further increase the number of synthetic identities. Code will be released at `https://github.com/Tencent/TFace`.

## 1 INTRODUCTION

Recent years have witnessed incredible improvements in the accuracy of FR models. This can be attributed to the advancements in model architectures (Boutros et al., 2022b; Huang et al., 2017), margin-based loss functions (Deng et al., 2019; Huang et al., 2020), and more importantly, the availability of large-scale face datasets (Huang et al., 2008; Kemelmacher-Shlizerman et al., 2016; Zhu et al., 2021), which contain millions of identities with rich variations in age, pose and expression.

However, these large-scale datasets are often collected directly from the Internet, without the explicit consent of individuals, which inevitably leads to privacy and legal issues (Regulation, 2016). Moreover, these datasets suffer from challenges about noisy labels (Wang et al., 2018a; 2019) and imbalanced class distribution (Liu et al., 2019; Yi et al., 2014). In other words, images with the same label may belong to different individuals, and there is a significant disparity in the number of images among different identities. These drawbacks limit the further application of real face datasets.

---

[*]Equal contribution
[†]Corresponding author

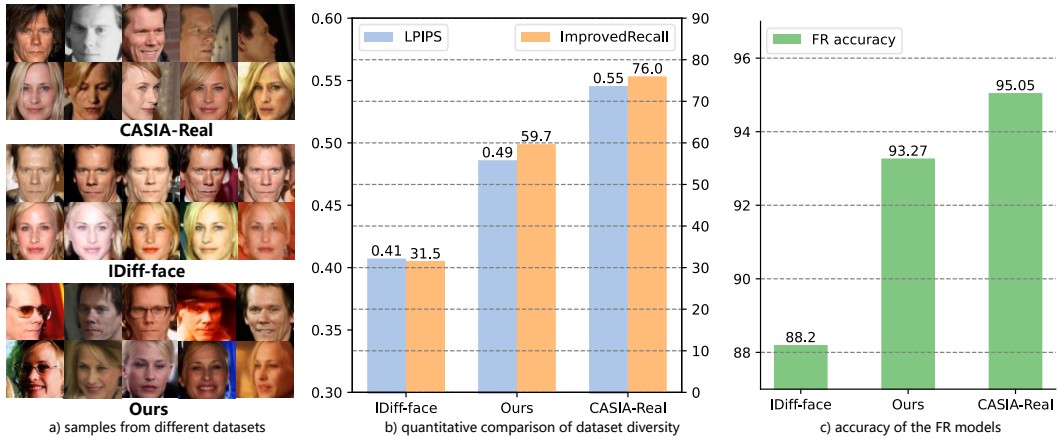

Figure 1: a) **Visualization of samples from different datasets.** Samples from the real face dataset CASIA exhibit variations in attributes such as pose, expression and illumination. However, when we synthesize face data using previous method IDiff-Face (Boutros et al., 2023), the generated images show poor diversity, more specifically, similar expressions and poses, which is caused by identity overfitting. In contrast, our method can generate a wider variety of images, thereby enhancing the accuracy of the trained FR model. b) **Quantitative comparison of dataset diversity.** We apply LPIPS and Improved Recall to measure the diversity of different datasets. A higher value indicates better diversity. c) **Quantitative comparison of final accuracy of the FR models.**

Generating synthetic data of non-existent identities to train FR model, also known as synthetic face recognition, is a feasible solution to the above issues as suggested by DeAndres-Tame et al. (2024). There have already been some explorations in this field. The initial methods (Shen et al., 2018; Deng et al., 2020) employ Generative Adversarial Networks (Goodfellow et al., 2020) to generate synthetic face data and utilize the disentangled latent space to enhance the controllability. However, GAN-based methods have been shown to generate only a limited number of unique identities (Kim et al., 2023), leading to poor generalization of the trained FR models. Recently, diffusion models have made significant advancements in image generation (Song et al., 2020a; Dhariwal & Nichol, 2021; Song et al., 2020b; Rombach et al., 2022). Some diffusion-based methods (Wang et al., 2024; Li et al., 2024b; Boutros et al., 2023; Papantoniou et al., 2024; Zhang et al., 2024) have been proposed for identity-preserving face generation, which is achieved by conditioning the generation process with identity contexts, i.e., identity features extracted from the pretrained FR models (Deng et al., 2019; Boutros et al., 2022a). It has been demonstrated that diffusion-based methods are capable of generating a greater variety of unique identities compared to GAN-based methods (Kim et al., 2023), showing a highly promising potential for real applications. Yet, a high-quality human face dataset does not just imply a large number of identities but also requires good intra-class diversity. Specifically, the synthetic face images need to exhibit variations in attributes such as expression, illumination and pose.

While existing methods have made significant progress, they still suffer from context overfitting (Boutros et al., 2023). More specifically, a fixed identity context not only determines the identity of generated images but also limits their the variations of identity-irrelevant attributes, resulting in insufficient intra-class diversity. As illustrated in Figure 1a, previous methods often exhibit a reduced diversity of synthetic images, i.e., lack of variations in expression and pose, compared to real-world dataset. To further demonstrate this issue, following Kim et al. (2023) and Papantoniou et al. (2024), we utilize LPIPS (Zhang et al., 2018) and Improved Recall (Kynkäänniemi et al., 2019) to quantitatively measure the intra-class diversity of these datasets. As shown in Figure 1b, the diversity of the synthetic dataset generated by previous approach significantly lags behind that of the real dataset CASIA-Webface (Yi et al., 2014), resulting in poor performance of the trained FR model (Figure 1c). Although methods such as Contextual Partial Dropout (Boutros et al., 2023), style condition extractor (Kim et al., 2023) and paired data generation (He et al., 2024) have been proposed to alleviate this issue, they either rely on complex network designs or require the introduction of additional training data, and still have a significant performance gap compared to real data-based methods.

Intuitively, the model inherently possesses the ability to generate diverse images because the real data used to train the generative model contain rich intra-class variations. However, when it comes to a specific identity context, such inherent capability is limited because of context overfitting. Therefore, in this paper, we propose a novel framework to unleash model inherent capability to enhance intra-class diversity for synthetic face recognition, shortened as **UIFace**. Specifically, we first train a diffusion model that can perform sampling conditioned on either specific identity contexts or a learnable empty context. When conditioned on a specific identity context, the model generates identity-preserving images but with poor diversity due to context overfitting. On the other hand, when conducting generation conditioned on the empty context, the model can fully leverage its inherent capability to synthesize various images but with random identities. This is because during training, we allow the empty context to generate all the images in the dataset. To exploit the strength of both types of contexts to synthesize images that are diverse as well as identity-preserving, we adopt a novel two-stage generation strategy during inference. Our key observation is that in the early stages of sampling, the model restores identity-irrelevant contents, while in the later stage the model recovers the identity-relevant details (Section 3.2). Thus, in the first stage, the diffusion model performs sampling conditioned on the empty context for unleashing its intrinsic ability to generate large intra-class variations. In the second stage, the model generates identity-preserving details based on given identity conditions. Such a two-stage strategy takes advantage of model's inherent ability to enhance the diversity of conditional generation. Moreover, we propose an adaptive partitioning strategy to adaptively determine the boundary of these two stages for each sample based on the difference between adjacent cross-attention maps. To fully harness the diversity of empty context-conditioned generation, we propose an attention injection module to use the attention maps from unconditional generation to guide the conditional generation process, which further leverages the model's inherent ability while maintaining ID-consistency.

In summary, our contributions are as follows.

- We propose a novel two-stage synthetic face recognition framework. By allowing the learnable empty context and identity contexts to dominate different stages of the face generation, our method can fully leverage model's inherent capability to achieve intra-class-diversified image generation while maintaining identity-preserving.

- We propose an adaptive partitioning strategy to adaptively determine the boundaries of two stages for different samples and an attention injection module to utilize attention maps from unconditional generation to guide conditional generation. These designs can further unleash model's inherent ability to enhance the intra-class variations of synthesized images.

- Experimental results show that our method outperforms existing state-of-the-art methods by a significant margin by using fewer training data. We surpass current state-of-the-art methods even when synthesizing fewer than half the number of synthetic identities. When further increasing the number of identities, the proposed UIFace can surprisingly achieve comparable performance with those FR models trained on real datasets.

## 2 RELATED WORK

### 2.1 FACE RECOGNITION

Face recognition involves identifying or verifying a person from an enrolled dataset. With the continuous improvement of network architectures (He et al., 2016; Boutros et al., 2022b; Huang et al., 2017) and the introduction of novel loss functions (Boutros et al., 2022a; Deng et al., 2019; Huang et al., 2020; Wang et al., 2018b; Kim et al., 2022), the accuracy of face recognition models has made remarkable advancements in recent years. More importantly, the improvement in performance is also attributed to large-scale face datasets (Huang et al., 2008; Cao et al., 2018; Zhu et al., 2021; Guo et al., 2016; Kemelmacher-Shlizerman et al., 2016) as well as datasets tailored to address specific challenges (Zheng & Deng, 2018; Zheng et al., 2017; Sengupta et al., 2016; Moschoglou et al., 2017). Nevertheless, these datasets are collected directly from the Internet without explicit individual consent, leading to inevitable privacy and legal concerns. Moreover, they are suffered from noisy labels and long-tail problem (Yi et al., 2014). The above issues need to be carefully considered before using these datasets.

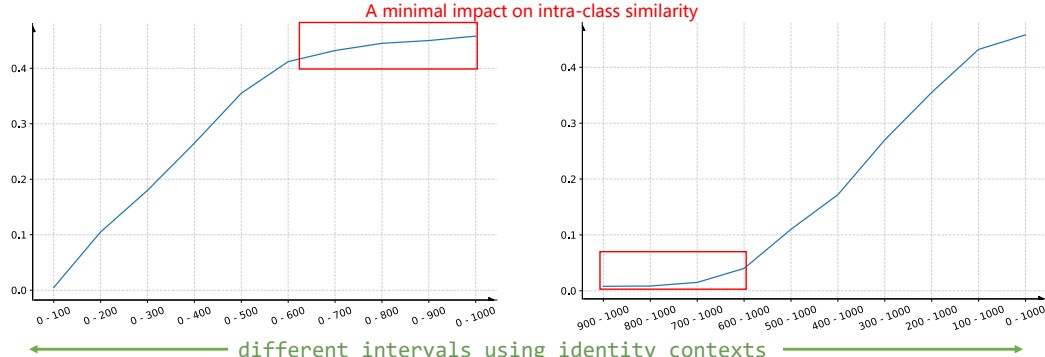

Figure 2: **Effects of different sampling timesteps on identity.** The x-axis represents timestep intervals where identity contexts are used as conditions. The empty context is used as a substitute in timesteps that not covered in intervals. The y-axis represents the intra-class similarity of the generated face images. The maximum sampling timestep $T$ is set to 1000.

## 2.2 FACE IMAGE SYNTHESIS

Image generation models have made remarkable progress, allowing the synthesis of high-quality human face images. GAN-based methods (Brock, 2018; Choi et al., 2018; Karras, 2017; Karras et al., 2019; 2020; Yin et al., 2017) are successful pioneers among them. They achieve identity-preserving face generation by decoupling identity and attributes (Bao et al., 2018), or introducing additional classifiers and discriminators (Shen et al., 2018). DiscoFaceGAN (Deng et al., 2020) introduces a more fine-grained decoupled latent space to enable precise control of synthesized faces. Recently, diffusion models (Song et al., 2020a; Ho et al., 2020; Dhariwal & Nichol, 2021; Song et al., 2020b; Rombach et al., 2022) have made significant advances in the field of image synthesis. Some methods (Zhang et al., 2024; Wang et al., 2024; Li et al., 2024b) have achieved high-fidelity identity-preserving image generation with text and ID-conditioned diffusion model.

## 2.3 FACE RECOGNITION WITH SYNTHETIC DATASET

Replacing real datasets with synthetic face datasets can address the legal and class imbalance issues. SynFace (Qiu et al., 2021) applies DiscoFaceGAN (Deng et al., 2020) to synthesis identity-consistent data for training FR models. DCFace (Kim et al., 2023) proposes an diffusion-based method that employs decoupled style and identity encoders to generate dual conditions and demonstrates that DDPM (Ho et al., 2020) can generate more unique identities than GAN-based methods, which is crucial for improving the accuracy of FR models. Some works (Boutros et al., 2023; Papantoniou et al., 2024) continue to enhance the quality of synthetic datasets, narrowing the performance gap between FR models trained on synthetic and real data. Nevertheless, as depicted in Figure 1, existing methods still suffer from context overfitting, resulting in insufficient diversity in synthetic datasets. While some approaches (Boutros et al., 2023; Kim et al., 2023; He et al., 2024) have been suggested to address this challenge, they often depend on intricate network architectures or necessitate extra training data. In contrast, this paper explores to unleash model's inherent capability to intra-class-diversified image generation for synthetic-based face recognition.

## 3 METHOD

**Overview.** The aim of synthetic face recognition is to generate synthetic face data for training FR models, thereby addressing the inherent issues of real datasets collected directly from the Internet, as discussed in Section 1. Formally, given the training dataset $\{x_i\}_{i=1}^N$, we follow previous works (Papantoniou et al., 2024; Boutros et al., 2023) to utilize a pre-trained FR model $\mathcal{F}$ to extract the identity context $c_i = \mathcal{F}(x_i)$ for each image, resulting in extended dataset $\{x_i, c_i\}_{i=1}^N$. Subsequently, we train a conditional diffusion model $\mathcal{G}$ that can perform face generation conditioned on either specific identity contexts $c_i$ or en empty context $c_e$ (Section 3.1). Then we elucidate the crucial

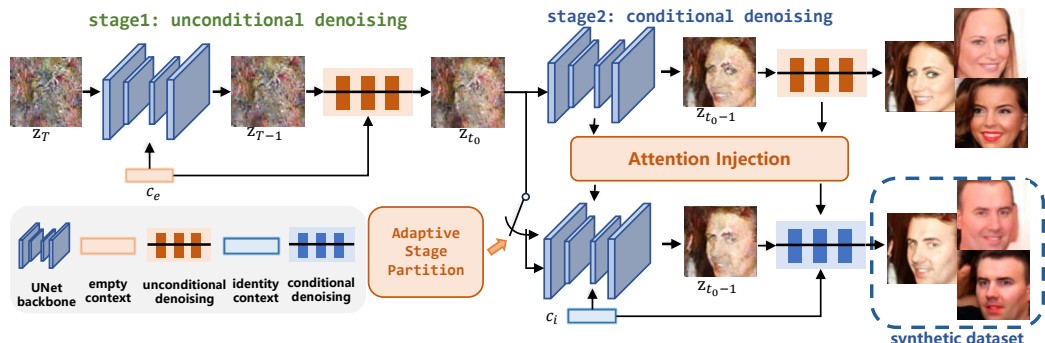

Figure 3: **Overview of proposed UIFace.** We propose a two-stage sampling strategy to unleash the intrinsic capability of the model to generate diverse images. In the first stage, the model performs unconditional generation based on the empty context $c_e$. In the second stage, the model restores identity-relevant details conditioned on specific identity contexts $c$. We further propose an adaptive stage partition strategy to determine the boundary of these two stages $t_0$ and an attention injection module to enhance diversity of synthetic dataset while maintaining identities.

observation that in the early sampling stage, the model restores identity-irrelevant contents, whereas in the later stage it recovers details that determine the identity (3.2). Next, we introduce the two-stage sampling strategy to generate synthetic face images that simultaneously preserve identity and exhibit diversity (Section 3.3). Then, we propose an attention injection mechanism to further enhance the quality of the synthetic images (Section 3.4). Lastly, we present details about synthetic dataset generation (Section 3.5). The framework overview is shown in Figure 3.

## 3.1 PRELIMINARY

Latent Diffusion Model (Rombach et al., 2022) employs a denoising process to approximate the distribution of latent representations $z$ of real images $x$. A Variational Autoencoder $\mathcal{E}$ is first applied to map the images to a lower-dimensional latent space $z = \mathcal{E}(x)$. The images are progressively corrupted by adding Gaussian noise according to a predefined schedule during training. Formally,

$$z_t = \sqrt{\bar{\alpha}_t} z_0 + \sqrt{1 - \bar{\alpha}_t} \epsilon, \tag{1}$$

where $t$ and $z_0$ stand for the diffusion timestep and clean latent representation respectively. The reverse process (also known as the sampling process) is defined by the following equation

$$z_{t-1} = \mu_\theta(z_t, t, c) = \frac{1}{\sqrt{\alpha_t}} \left( z_t - \frac{1 - \alpha_t}{\sqrt{1 - \bar{\alpha}_t}} \epsilon_\theta(z_t, t, c) \right). \tag{2}$$

where $c$ is the textual condition. The optimization of the UNet (Ronneberger et al., 2015) backbone $\epsilon_\theta$ is achieved by recovering the random noise $\epsilon$ and minimizing the following loss

$$\mathcal{L} = \mathbb{E}_{z_t, t, c, \epsilon \sim \mathcal{N}(0, \mathbf{I})} \left[ \| \epsilon - \epsilon_\theta(z_t, t, c) \|_2^2 \right]. \tag{3}$$

In synthetic-based face recognition, $c$ is the identity context extracted from a pre-trained FR model. Our method introduce an additional learnable empty context $c_e$. During training iterations, the identity context $c$ is probabilistically replaced with $c_e$. Thus, $c_e$ can be used to generate images of any identities present in the training data. When training finished, the model can generate identity-preserving images conditioned on any given $c$, or random face images conditioned on $c_e$. Our training process is consistent with previous ID-conditioned diffusion methods except for the introduction of empty context. Next, we will describe the inference process of our method.

## 3.2 EFFECTS OF DIFFERENT SAMPLING TIMESTEPS ON IDENTITY

In this subsection, we present our observations about effects of different sampling timesteps on the identity of generated images. For this purpose, we inject identity contexts as generation conditions

in different timestep intervals and calculate the intra-class similarity of generated images, which is based on the distance of features from a pre-trained FR model just as the standard metric in this field. As depicted in Figure 2, during the initial generation phase (for larger $t$ values), employing identity contexts as conditions exhibits minimal enhancement in the intra-class similarity of synthesized images, while condition the generation process in the later timesteps (for smaller $t$ values) can rapidly improve the intra-class similarity. It implies that the model recovers identity-irrelevant contents at early sampling stage and restores identity-relevant details in later stage. Based on these observations, we propose a two-stage sampling strategy to unleash the inherent diversity of the model while maintaining the identity consistency of generated images.

### 3.3 TWO-STAGE SAMPLING STRATEGY

As mentioned in the Section 1, previous synthetic-based methods suffer from the overfitting of identity context $c$, leading to poor diversity of generated face images and degenerated face recognition performance. To address this issue, we propose a two-stage sampling strategy during inference to unleash the model's inherent capability to enhance intra-class diversity. According to the observations from Section 3.2, we instruct the model to perform sampling conditioned on the empty identity $c_e$ in the early stage, which helps to introduce greater intra-class variations. In the later stage, we condition the generation process with specific identity context $c$ to generate identity-relevant details. This strategy leads to both diverse and identity-preserving images. The overall formula is as follows,

$$
z_{t-1} = \begin{cases} \frac{1}{\sqrt{\alpha_t}} \left( z_t - \frac{1-\alpha_t}{\sqrt{1-\bar{\alpha}_t}} \epsilon_\theta(z_t, t, c_e) \right) & \text{if } t \in [t_0, T], \\ \frac{1}{\sqrt{\alpha_t}} \left( z_t - \frac{1-\alpha_t}{\sqrt{1-\bar{\alpha}_t}} \left( (w+1)\epsilon_\theta(z_t, t, c) - w\epsilon_\theta(z_t, t, c_e) \right) \right) & \text{if } t \in [0, t_0], \end{cases} \tag{4}
$$

where $t_0$ serves as the boundary between the two stages during inference and $w$ is the scale of classifier-free guidance (Ho & Salimans, 2022). Next we further introduce an adaptive partitioning strategy to assign a unique boundary $t_0$ for each sample.

**Adaptive partitioning.** We observed that the cross-attention maps between UNet features and identity context exhibit significant variations during the early sampling timesteps when the model has not yet started focusing on details that determine the identity. And a decrease of temporal difference in cross-attention maps signifies the focus of sampling process shifting towards identity-relevant details in images (See Appendix for further details). Thus we propose an adaptive partition strategy based on the temporal differences of cross-attention maps. Specifically, let $h_t$ denote the normalized cross-attention map between the learnable empty context $c_e$ and the UNet features at timestep $t$ and $d_t$ denote the difference of adjacent cross-attention maps. The boundary $t_0$ for each sample is determined as follows,

$$
d_t = \|h_{t+1} - h_t\|_2, \quad t_0 = \min\{t : (d_{t+1} > th) \text{ and } (d_t < th)\} \tag{5}
$$

where $th$ is a hyperparameter. We interpolate the cross-attention maps of each layer to the same resolution, then average and normalize them across the spatial dimensions to get the final $h_t$. The attention maps mentioned later in this paper are obtained in a similar manner.

### 3.4 ATTENTION INJECTION

To further enhance the quality of synthesized images by leveraging the inherent diversity of the empty context $c_e$, we propose an attention injection module in the second sampling stage to directly use the attention maps of $c_e$ to guide the conditional generation. Let $h_{self}^c$ and $h_{cross}^c$ represent the self-attention map of UNet features in conditional sampling and cross-attention map between UNet features and identity context $c$ respectively. Similarly, let $h_{self}^{c_e}$ and $h_{cross}^{c_e}$ denote the attention maps in unconditional generation. A naive approach is to directly copy the corresponding attention maps. Nevertheless, experimental results indicate that directly substituting the cross-attention map has a considerable impact on identity and quality of the generated images (Section 4.3). Thus, we introduce a novel injection strategy to leverage the cross-attention map $h_{cross}^{c_e}$ from $c_e$. Specifically, we normalize $h_{cross}^c$ using the mean and variance of $h_{cross}^{c_e}$ along the spatial dimensions. The formula is as follows,

$$
\mu_c, \sigma_c = \text{mean}(h_{cross}^c), \ \text{std}(h_{cross}^c), \quad \mu_{c_e}, \sigma_{c_e} = \text{mean}(h_{cross}^{c_e}), \ \text{std}(h_{cross}^{c_e}), \tag{6}
$$

$$
h_{cross}^c = \frac{h_{cross}^c - \mu_c}{\sigma_c} * \sigma_{c_e} + \mu_{c_e}. \tag{7}
$$

Table 1: **Comparisons with state-of-the-art synthetic-based face recognition methods.** Face verification accuracy (%) on difference benchmarks.

| Method | Num of imgs (IDs × imgs/ID) | LFW | CFP-FP | CPLFW | AGEDB | CALFW | Average |
|---|---|---|---|---|---|---|---|
| CASIA-Real | ~0.5M( 10.5K × 47) | 99.43 | 97.27 | 90.18 | 94.78 | 93.58 | 95.05 |
| SynFace | 0.5M(10k × 50) | 91.93 | 75.03 | 70.43 | 61.63 | 74.73 | 74.75 |
| DigiFace | 0.5M(10k × 50) | 95.4 | 87.4 | 78.87 | 76.97 | 78.62 | 83.45 |
| DCFace | 0.5M(10k × 50) | 98.55 | 85.33 | 82.62 | 89.70 | 91.60 | 89.56 |
| IDiff-Face | 0.5M(10k × 50) | 98.0 | 85.47 | 80.45 | 86.43 | 90.65 | 88.20 |
| $ID^3$ | 0.5M(10k × 50) | 97.68 | 86.84 | 82.77 | 91.0 | 90.37 | 89.80 |
| CemiFace | 0.5M(10k × 50) | 99.03 | 91.06 | 87.65 | **91.33** | 92.42 | 92.30 |
| Arc2Face | 0.5M(10k × 50) | 98.81 | 91.87 | 85.16 | 90.18 | **92.63** | 91.73 |
| UIFace | 0.5M(10k × 50) | **99.27** | **94.29** | **89.58** | 90.95 | 92.25 | **93.27** |
| DigiFace | 1.2M(10k × 72 + 100k × 5) | 96.17 | 89.81 | 82.23 | 81.10 | 82.55 | 86.37 |
| DCFace | 1.2M(20k × 50 + 40k × 5) | 98.58 | 88.61 | 85.07 | 90.97 | 92.82 | 91.21 |
| CemiFace | 1.0M(20k × 50) | 99.18 | 92.75 | 88.42 | 91.97 | 93.01 | 93.07 |
| Arc2Face | 1.2M(20k × 50 + 40k × 5) | 98.92 | 94.58 | 86.45 | **92.45** | **93.33** | 93.14 |
| UIFace | 1.0M(20k × 50) | **99.22** | **95.03** | **90.42** | **92.45** | 93.18 | **94.06** |
| UIFace | 1.5M(30k × 50) | 99.38 | 95.96 | 90.67 | 93.28 | 93.43 | 94.54 |

As for the self-attention map, we adopt direct replacement as $h^c_{self} = h^{c_e}_{self}$ because we found that the self-attention of UNet features affects identity-irrelevant attributes of the synthesized images. Experimental results demonstrate that the proposed attention injection module, which handles self and cross-attention differently, achieves identity preservation while further harnessing the inherent ability of diffusion model to enhance the diversity of the synthesized images.

## 3.5 SYNTHETIC-BASED FACE RECOGNITION

To generate synthetic dataset with non-existent identities, first we follow the previous method (Boutros et al., 2023) to generate non-existent face images using an additional unconditional face generation model and extract their identity contexts using the pre-trained FR model (Boutros et al., 2022a). Then we adopt the strategy from Kim et al. (2023) to filter out similar identity contexts using cosine distances to ensure inter-class discrepancy. Subsequently, we utilize these identity contexts as conditions to generate face dataset using our method. Last, we train new FR models on these synthetic data from scratch and report the final FR performance on real datasets.

## 4 EXPERIMENTS

### 4.1 EXPERIMENTAL SETUP

**Training and testing dataset.** We train our diffusion model on CAISA-Webface (Yi et al., 2014) dataset containing about 500k face images of 10575 real identities of celebrities directly grabbed from the web, which show large variations of attributes such as pose, illumination, and facial expressions. We evaluate the final synthetic-based FR models on the five most commonly used real datasets, LFW (Huang et al., 2008), CFP-FP (Sengupta et al., 2016), CPLFW (Zheng & Deng, 2018), AgeDB (Moschoglou et al., 2017) and CALFW (Zheng et al., 2017). These datasets include images with varying ages, poses, and facial expressions, which allow to comprehensively measure the generalization of FR models.

**Implementation details.** Our diffusion model is trained using an Adam (Kingma, 2014) optimizer with a learning rate of 1e-4 and a batch size of 64. The training is conducted for a total of 250k iterations. During training process, we randomly replaced $c$ with the empty context $c_e$ at a probability of 20%. As for training synthetic-based FR models, We adopt an IR50 as backbone with ArcFace loss (Deng et al., 2019) for 40 epochs. The scale of classifier-free guidance is set to 1 and $th$ is 0.005.

### 4.2 COMPARISON WITH STATE-OF-THE-ART METHODS

In this subsection, we compare our method with other synthetic-based face recognition methods. The results are shown in Table 1. As shown in the table, our method significantly outperforms previous approaches, including the latest methods (Papantoniou et al., 2024; Sun et al., 2024; Li et al.,

Table 2: **Ablation studies.** Face verification accuracy (%) on difference benchmarks and diversity metrics of synthetic datasets from diffent settings.

| Method | LPIPS | ImRecall | LFW | CFP-FP | CPLFW | AGEDB | CALFW | Average |
|---|---|---|---|---|---|---|---|---|
| baseline | 0.5270 | 53.99 | 98.98 | 92.57 | 87.0 | 88.42 | 90.7 | 91.53 |
| baseline + 2-stage-fixed | 0.5302 | 57.92 | 98.83 | 92.94 | 88.37 | 89.1 | 91.15 | 92.08 |
| baseline + 2-stage-adaptive | 0.5346 | 62.35 | 99.05 | 94.13 | 88.93 | 89.92 | 91.48 | 92.70 |
| baseline + attn | 0.5338 | 61.98 | 99.17 | 92.73 | 88.12 | 89.47 | 91.75 | 92.25 |
| baseline + 2-stage-fixed + attn | 0.5367 | 63.77 | 99.15 | 93.84 | 88.48 | 90.3 | 91.78 | 92.71 |
| baseline + 2-stage-adaptive + attn | **0.5592** | **71.96** | **99.27** | **94.29** | **89.58** | **90.95** | **92.25** | **93.27** |

2024a) in the average metrics across the five datasets. Moreover, our method maintains superior performance even when synthesizing fewer than half the number of face images compared to previous methods. Notably, previous state-of-the-art method Arc2Face (Papantoniou et al., 2024) even uses more datasets and higher resolution images to train the generative model. We attribute the superiority of our method to our two-stage sampling strategy and attention injection mechanism, which can unleash the potential of diffusion model to synthesize images with enhanced diversity, thereby enabling the FR model to have better generalization capabilities on real datasets. By increasing the number of identities, our approach even outperforms the FR model trained on CAISA-Webface on the CPLFW dataset and achieves comparable performance in the average accuracy.

## 4.3 ABLATION STUDIES

In this subsection we further investigate the effectiveness of each design of proposed **UIFace** in synthetic face recognition. The results are shown in Table 2.

**Impact of two-stage sampling.** We first build a baseline method based on the vanilla single-stage sampling strategy (baseline) for synthetic-based face recognition. To validate the effectiveness of proposed two-stage sampling, we first employ a naive two-stage sampling process based on fixed stage partition ($t_0 = 500$, 2-stage-fixed). As shown in Table 2, the naive two-stage design with fixed partition has already demonstrated a significant performance improvement compared to the baseline in both the diversity of the generated dataset and the accuracy of the FR model. We attribute this to our observations in Section 3.2 so that such two-stage strategy can unleash the model's potential for intra-class-diversified image generation, leading to improved face recognition accuracy.

**Impact of adaptive partitioning strategy.** Moreover, we validate the effectiveness of the proposed adaptive partition by incorporating it into the baseline method (2-stage-adaptive). As shown in Table 2, the proposed adaptive partitioning significantly outperforms the fixed partitioning method. Such adaptive partition strategy not only eliminates the need of manual hyperparameter tuning but also allows for the allocation of different stage boundaries for each sample, especially considering that the partition preferences may vary across different samples.

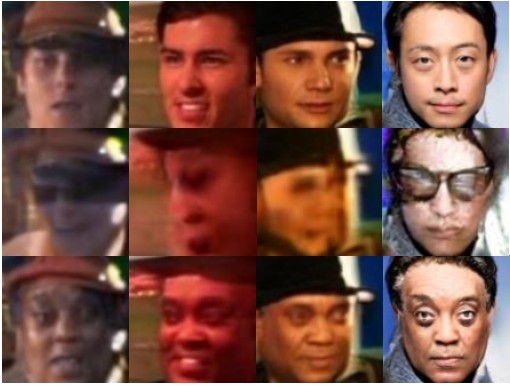 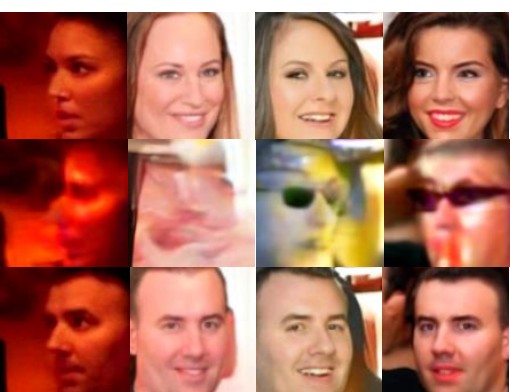

Figure 4: visualization results of different attention injection strategies (**Top**: unconditional generation; **Middle**: attention injection with vanilla replacement; **Bottom**: the proposed attention inject).

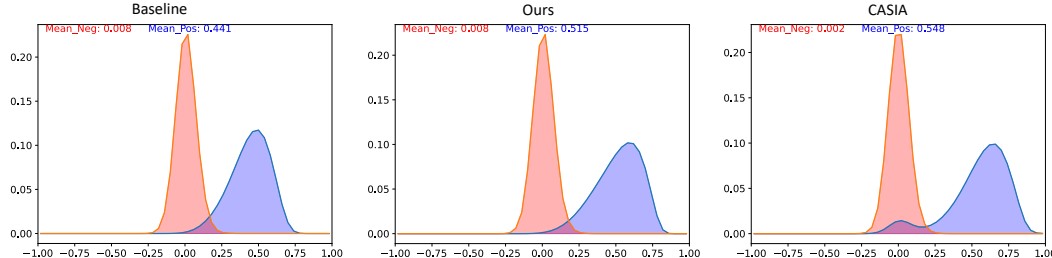

Figure 5: Genuine and imposter comparisons. (**Left**: baseline; **Mid**: UIFace; **Right**: CASIA).

Table 3: **UIFace with different diffusion sampling methods.** Our method is agnostic to any specific diffusion sampling method, and achieves performance improvements across different methods.

| Method | Sampling | LFW | CFP-FP | CPLFW | AGEDB | CALFW | Average |
|--------|----------|-----|--------|-------|-------|-------|---------|
| baseline | DDIM | 98.98 | 92.57 | 87.00 | 88.42 | 90.70 | 91.53 |
| UIFace | DDIM | **99.27** | **94.29** | **89.58** | **90.95** | **92.25** | **93.27** |
| baseline | FPNDM | 99.18 | 93.03 | 87.70 | 88.92 | 91.18 | 92.00 |
| UIFace | FPNDM | **99.20** | **94.66** | **89.75** | **91.5** | **92.86** | **93.59** |

**Impact of the proposed attention injection.** We first visualize the synthesized images using the proposed attention injection compared to images generated using vanilla attention map replacement. As shown in Figure 4, directly replacing the attention map will greatly degenerate the quality and identity of the synthesized images, while the proposed attention injection can effectively utilize the diversity from unconditional generation as well as maintain identity-preserving. Then we augment the baseline model with the proposed attention injection method (attn). A significant improvement can be observed in Figure 2, in both the diversity of synthetic datasets and the accuracy of the FR model. This is because we enhance the intra-class variations with the model's inherent ability in unconditional diverse image generation. Combining our adaptive two-stage strategy and attention injection leads to the best diversity and performance improvement.

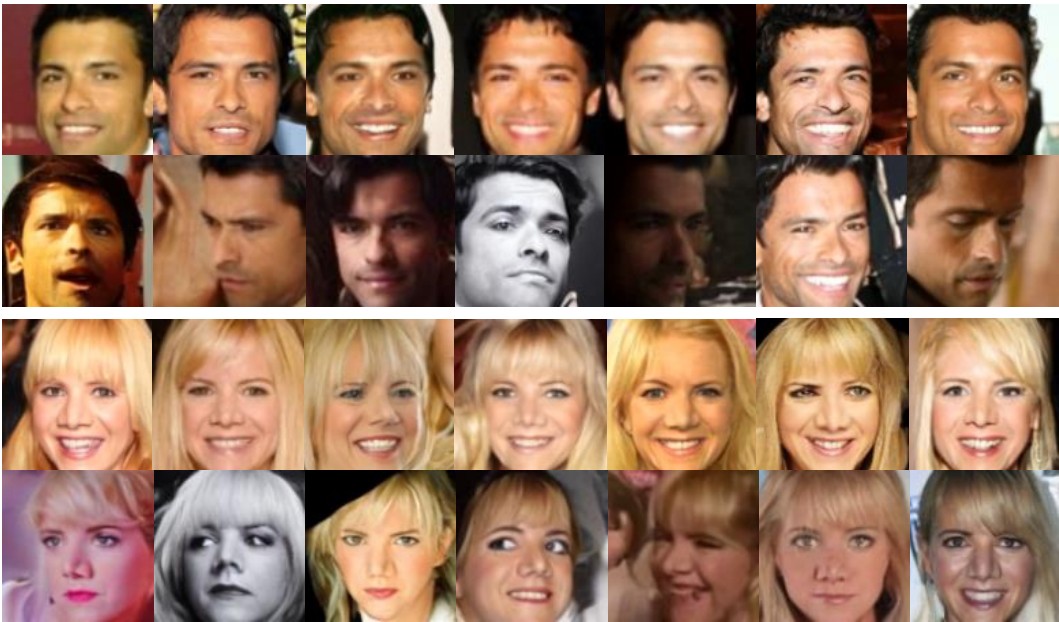

Figure 6: Visualization results of IDiff-Face (**odd rows**) and UIFace (**even rows**).

**Discussion about identity consistency.** Intuitively, the two-stage sampling strategy proposed in this paper may reduce the intra-class identity similarity of the generated images since the empty context $c_e$ stands for "random identities", which is also of great significance in synthetic face recognition. In fact, there is a trade-off between intra-class diversity and intra-class identity consistency in this task. And our designs, including the discussion in Section 3.2, the use of classifier-free guidance in the second stage (Section 3.3) and the different treatment of self and cross-attention maps in Section 3.4, have sought to maximize intra-class diversity while preserving identity consistency as much as possible. To demonstrate this, we conduct genuine and imposter comparisons, a common metric in face recognition that measures the similarity between data points of the same individual and different individuals. As shown in Figure 5 and Table 2, our method has improved both intra-class identity similarity and style diversity compared to the baseline method.

### 4.4 UIFACE WITH DIFFERENT DIFFUSION SAMPLING METHODS

Since our motivation lies in different sampling preferences in different stages during inference, which is agnostic to any specific diffusion sampling method, our method can be applied to various diffusion-based generation methods. Here we choose FPNDM (Liu et al., 2022) as a representative. As demonstrated in Table3, UIFace achieves consistent improvements on both DDIM and FPNDM sampling methods, showing the effectiveness and generalization of our method. The baseline is the same as Section4.3 and the numbers of synthetic images are all 0.5M.

### 4.5 QUALITATIVE RESULTS

The qualitative results of IDiff-Face (Boutros et al., 2023) and proposed UIFace are shown in Figure 6. We randomly sample identity contexts and used both methods to generate synthetic face images. It is evident from the figure that the images generated by IDiff-Face exhibit a lack of variations in attributes such as expression, illumination and pose, which is caused by context overfitting as discussed in Section 1. In contrast, our proposed **UIFace** employs a two-stage sampling strategy to unleash the model's inherent capability, thereby enhancing intra-class diversity in the generated images (more expression changes, facial rotations and lighting variations).

## 5 CONCLUSION

We introduce **UIFace**, a novel synthetic face recognition framework that leverages a two-stage sampling strategy to unleash inherent model capability to enhance intra-class diversity of synthetic face dataset. Moreover, we propose an adaptive partitioning strategy and an attention injection method to further improve intra-class diversity while maintaining identity-preserving. Extensive experiments demonstrate that **UIFace** outperforms existing methods in multiple benchmarks even using less training data and fewer identity number.

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
