# UIFace: Unleashing Inherent Model Capability to Enhance Intra-Class Diversity in Synthetic Face Recognition

**Xiao Lin**[1,*]**, Yuge Huang**[1,*]**, Jianqing Xu**[1]**, Yuxi Mi**[2]**, Shuigeng Zhou**[2]**, Shouhong Ding**[1,†]
[1]Tencent Youtu Lab    [2]Fudan University
{xiaolin,yugehuang,joejqxu,ericshding}@tencent.com
{yxmi20,sgzhou}@fudan.edu.cn

## A  Appendix

### A.1  More implementation details

**For the generative model**, we first use a pretrianed autoencoder VQGAN (Esser et al., 2021) from official repository of Stable Diffusion (Rombach et al., 2022) to map the input images to latent space of $3 \times 32 \times 32$. Then a UNet backbone with four resolution levels is implemented to predict the groundtruth noise in latent space. During trainig iterations, we apply a cossine annealing learning rate scheduler (Loshchilov & Hutter, 2016) in PyTorch (Paszke et al., 2019) and maintain an Exponential Moving Average (EMA) model with a momentum of 0.999 as the final generative model. As for reverse process, we use DDIM (Song et al., 2020) to accelerate the sampling process with a skip step of 20.

**For the recognition model**, our implementation is based on the official repository of TFace (`https://github.com/Tencent/TFace`) and IDiff-Face (Boutros et al., 2023). During training, we use the Adam optimizer and a step-wise descending learning rate schedule of $[0.1, 0.01, 0.001, 0.0001]$. We also apply data augmentation strategy from AdaFace (Kim et al., 2022) with a probability of 0.2.

**For evaluation**, we use a pretrained inception model (Szegedy et al., 2016) to extract embeddings of synthetic images to calculate ImprovedRecall (Kynkäänniemi et al., 2019) and a VGG-Net (Simonyan & Zisserman, 2014) to calculate LPIPS (Zhang et al., 2018) in this paper. For ImprovedRecall, we randomly sampled $10k \times 50$ images from the same 10k identities, with a nearest neighbor parameter $K$ set to 10. For LPIPS, we compute the average intra-class similarity of images from 100 randomly sampled identities.

### A.2  Performance gap between methods synthetic-based and real dataset-based methods.

Although synthetic-based face recognition methods can circumvent some issues about privacy, legality, and class imbalance, the performance gap between synthetic-based and real data-based methods exists due to distribution differences of real and synthetic datasets. We show this performance gap in Table 2. As shown in the table, our method achieves results closest to real data-based FR model, even with just half the size of the synthetic dataset compared to previous methods. When the number of synthetic identities is further increased to 20k, we even achieve competitive face recognition accuracy (~1%) against the real-based method.

### A.3  2-stage-fixed with different hyperparameter $t_0$

In method, we show that our adaptive partitioning strategy outperforms fixed partition strategy. Here, we proceed to present the experimental results under different settings of the hyperparameter $t_0$. The results are shown in Table1. It can be observed that for the fixed strategy, as $t_0$ decreases, the accuracy of the final FR model continues to increase. However, when is less than 500, the training of the recognition model collapses (random guess50%). This is because a smaller $t_0$ implies a longer first stage in the sampling process (unconditional generation), which enhances diversity but

Table 1: **Ablation experiments on the hyperparameter $t_0$**

| Method | $t_0$ | LFW | CFP-FP | CPLFW | AGEDB | CALFW | Average |
|---|---|---|---|---|---|---|---|
| baseline + 2-stage-fixed + attn | 400 | 50 | 50 | 50 | 50 | 50 | 50 |
| baseline + 2-stage-fixed + attn | 500 | 99.15 | 93.84 | 88.48 | 90.3 | 91.78 | 92.71 |
| baseline + 2-stage-fixed + attn | 600 | 99.07 | 93.49 | 88.2 | 89.7 | 91.05 | 92.30 |
| baseline + 2-stage-fixed + attn | 700 | 98.92 | 92.94 | 88.13 | 88.98 | 90.77 | 91.95 |
| baseline + 2-stage-adaptive + attn | | **99.27** | **94.29** | **89.58** | **90.95** | **92.25** | **93.27** |

decreases the intra-class consistency (discussed in ablation study). The diversity helps improve the recognition performance. However, when $t_0$ is too small, the intra-class consistency of the generated dataset is insufficient, resulting in training collapse. In contrast, our adaptive strategy outperforms all settings of fixed strategy, without the need to manually select the optimal $t_0$.

## A.4  ADDITIONAL ANALYSIS ABOUT $d_t$

Our adaptive partition strategy is based on temporal difference of cross-attention maps $\{d_t = h_{t+1} - h_t\}$. As shown in Figure 1, $\{d_t\}$ stays high values during the early stage of denoising, which implies that cross-attention maps change rapidly and model restores those identity-irrelevant contents such as facial rotations, illumination and backgrounds at the first stage. Then only after the cross-attention maps remain stable (low $d_t$ values) does the model begin to recover those identity-related details (as shown in Figure 1 right). These observations illustrate our motivation why we adopt the adaptive partition strategy based on $d_t$.

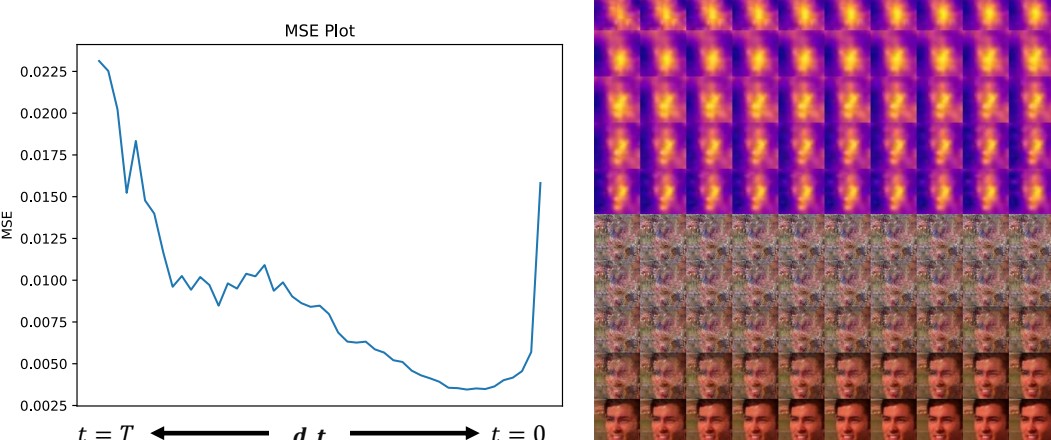

Figure 1: **Left**: $d_t$ plot. **Right**: Visualization of images and cross-attention maps during denoising process.

## A.5  MORE QUALITATIVE RESULTS

We provide a more visualization comparison between IDiff-Face and our method in Figure 2.

Table 2: **Comparisons with state-of-the-art synthetic-based face recognition methods on Real-Syn performance gap.** We calculate performance gap between synthetic-based and real dataset-based methods as (REAL - SYN)/SYN.

| Method | Num of imgs (IDs × imgs/ID) | Average | Performance gap |
|---|---|---|---|
| CASIA-Real | ~0.5M( 10.5K × 47) | 95.05 | **0.0%** |
| SynFace | 0.5M(10k × 50) | 74.75 | 27.2% |
| DigiFace | 0.5M(10k × 50) | 83.45 | 13.9% |
| DCFace | 0.5M(10k × 50) | 89.56 | 6.1% |
| IDiff-Face | 0.5M(10k × 50) | 88.20 | 7.8% |
| Arc2Face | 0.5M(10k × 50) | 91.73 | 3.6% |
| UIFace (ours) | 0.5M(10k × 50) | **93.27** | **1.9%** |
| DigiFace | 1.2M(10k × 72 + 100k × 5) | 86.37 | 10.0% |
| DCFace | 1.2M(20k × 50 + 40k × 5) | 91.21 | 4.2% |
| Arc2Face | 1.2M(20k × 50 + 40k × 5) | 93.14 | 2.0% |
| UIFace (ours) | 1.0M(20k × 50) | **94.06** | **1.1%** |
| UIFace (ours) | 1.5M(30k × 50) | **94.54** | **0.5%** |

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

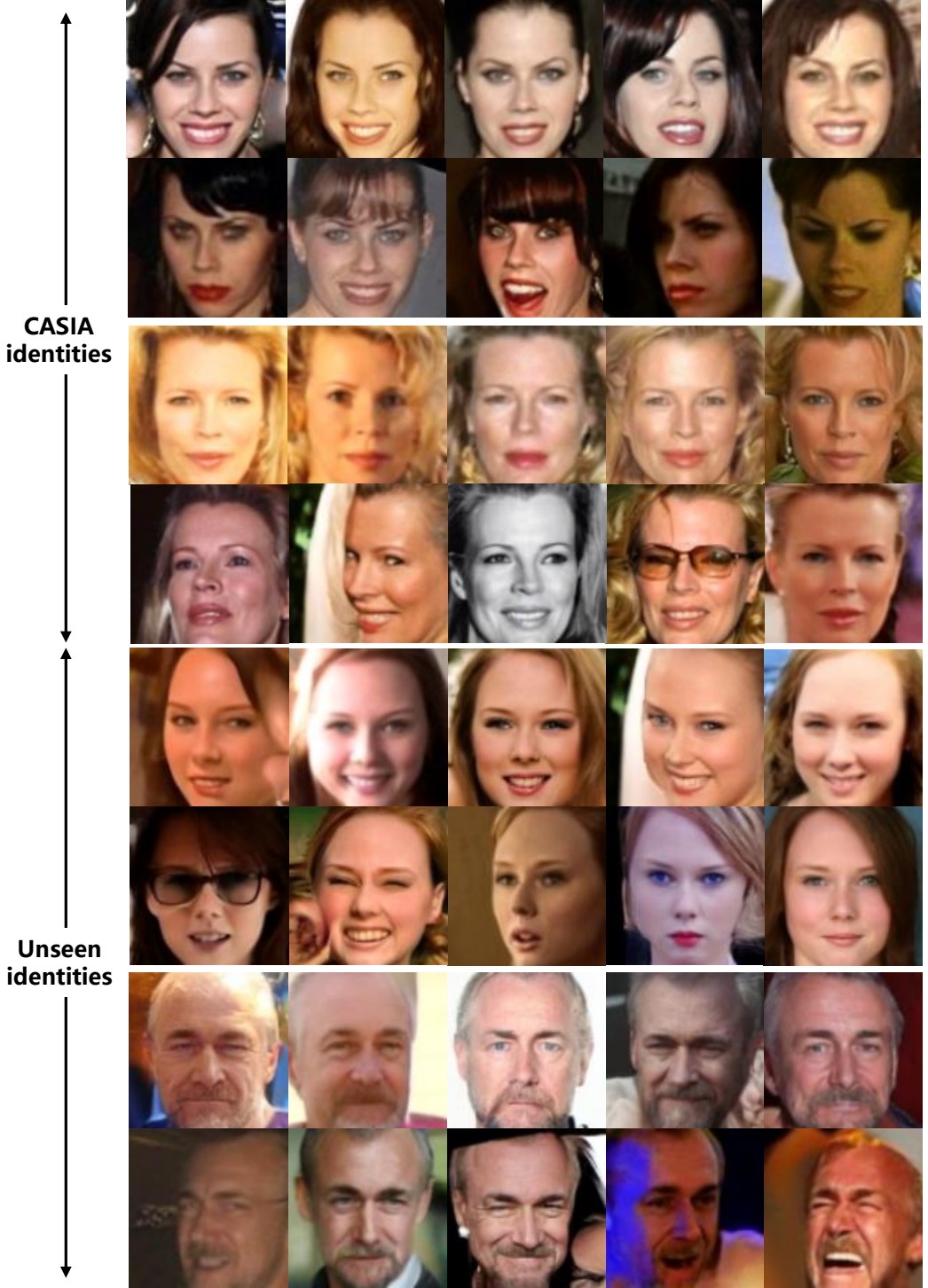

Figure 2: More visualization results of IDiff-Face (**odd rows**) and our UIFace (**even rows**) using either CASIA-Webface identity contexts or unseen identity contexts.