# OpenReview forum: "UIFace: Unleashing Inherent Model Capabilities to Enhance Intra-Class Diversity in Synthetic Face Recognition"
_ICLR.cc/2025/Conference — ICLR 2025 Poster_

### Official Review · Reviewer_zDSW · 2024-11-03

**Soundness:** 3
**Presentation:** 3
**Contribution:** 3
**Rating:** 6
**Confidence:** 4

**Summary:**

The paper addresses the challenge of generating diverse synthetic face images for the purpose of training face recognition models. The authors introduce a novel framework named UIFace, which employs a two-stage denoising approach alongside an attention injection module. This combination is designed to increase intra-class diversity while preserving identity consistency. As a result, UIFace achieves performance that is competitive with models trained on real-world datasets.

**Strengths:**

1. The two-stage denoising strategy and the attention injection module work together to generate diverse images while maintaining identity preservation, effectively addressing the issue of context overfitting.
2. The adaptive partitioning strategy dynamically identifies the optimal moment to transition between the two denoising stages, thereby enhancing both the quality and diversity of the generated images.
3. UIFace demonstrates its effectiveness by achieving performance on par with models trained on real datasets.

**Weaknesses:**

1. What about the scalability of the model? Are there any potential benefits to further increasing the number of identities beyond the 20K shown in Table 1?
2. While this paper focuses on enhancing the intra-class diversity for a specific identity, what about the inter-class discrepancies, which are also critical for training an effective face recognition model? How do the authors ensure this aspect is adequately addressed in their work?
3. The details of the overall training pipeline could be more clearly explained. For instance, in Stage 2, how are the losses from the unconditional and conditional branches calculated and combined?

**Questions:**

See weaknesses.

---

> ### Author Response · Authors · 2024-11-21
> **Response to Reviewer zDSW [1/1]**
>
> We are glad that Reviewer zDSW appreciates our work in terms of novelty and effectiveness. Here are responses to the questions you raised, hoping to address your concerns.
>
> **W1**: Thanks for your suggestion. Here we include additional experiment involving 30K identities * 50 images/identity, total 1.5M images. It is worth noting that the most recent works utilize maximum ~1M synthetic images to train FR models and our method achieves comparable performance to their ~1M datasets with just 0.5 million images. As the number of identities increases from 0.5M to 1.5M, the performance of UIFace exhibits continuous improvement, demonstrating the scalability of our method.
>
> | img_num | LFW   | CFP-FP | CPLFW | AGEDB | CALFW | Average |
> |---------|-------|--------|-------|-------|-------|---------|
> | 0.5M    | 99.27 | 94.29  | 89.58 | 90.95 | 92.25 | 93.27   |
> | 1M      | 99.22 | 95.03  | 90.42 | 92.45 | 93.18 | 94.06   |
> | 1.5M    | 99.38 | 95.96  | 90.67 | 93.28 | 93.43 | 94.54   |
>
> **W2**: Thank you for noticing this point. Your view about the factors that can have an impact on synthetic-based FR is very accurate and comprehensive, i.e, both intra-class diversity and inter-class discrepancy are indispensable.
>
> Since our paper mainly aims to address intra-class diversity, we directly adopt the strategy from DCFace [1] to ensure inter-class discrepancy. Specifically, we filter out similar synthetic identities by calculating cosine distances using a pretrained FR model and then use these refined non-existent identities to generate synthetic datasets in section 3.5. Such strategy from DCFace guarantees the discrepancy among synthetic identities in our paper.
>
> We will include this explanation in revised version.
>
> ```
> [1] Kim M, Liu F, Jain A, et al. Dcface: Synthetic face generation with dual condition diffusion model[C]//Proceedings of the ieee/cvf conference on computer vision and pattern recognition. 2023: 12715-12725.
> ```
>
> **W3**: First and foremost, we would like to clarify a few points about our method for better understanding.
>
> Our training process closely adheres to previous DDPM-based methods such as IDiff-Face [1]. The key improvement in training phase is the introduction of an additional empty context $c_e$. And the loss is consistent (equation 3) for both conditioned and unconditioned denoise.
>
> To sum up, the training is one-stage, and the loss is unified for both conditioned and uncondtioned denoise, same as vanilla DDPM. The difference lies in the conditions, which can be identity contexts or the empty context. And the two-stage strategy in our paper refer to sampling stage, not training stage.
>
> Perhaps some of our expressions such as "denosing stage", which can be used for both training and sampling, lead to misunderstandings. We will revise the phrasing and enhance readability in later version.
> ```
> [1] Boutros F, Grebe J H, Kuijper A, et al. Idiff-face: Synthetic-based face recognition through fizzy identity-conditioned diffusion model[C]//Proceedings of the IEEE/CVF International Conference on Computer Vision. 2023: 19650-19661.
> ```

---

### Official Review · Reviewer_JbLs · 2024-11-04

**Soundness:** 3
**Presentation:** 3
**Contribution:** 3
**Rating:** 6
**Confidence:** 4

**Summary:**

This paper tries to solve the problem of intra-class diversity with the generation of synthetic images in face recognition, and proposes a two-stage pipeline framework of DDPM with two improvements, including adaptive stage partition strategy and attention injection module. Solid results are achieved.

**Strengths:**

Originality: The method was proposed upon the previous DDPM with two tricks raised by the observation that early stage and later stage of denoising step can capture different semantic meanings of images.
Quality: The methodology is conceptually simple with massive experiments supported. The solid results have achieved on public benchmarks.
Clarify: The idea, albeit simple, has been clearly presented. It is better to re-structure introduce and related work slightly by minimizing the depiction of FR while maximizing the work related to data synthesis by generation methods.
Significance: The authors declared that the proposed methods achieved sota performance of face recognition by training models with synthetic images. There is no future work being declared to improve the work.

**Weaknesses:**

1. The method is proposed upon the previous DDPM work by using identity context maps as conditions. Two improvements, including adaptive partitioning strategy and attention injection module, have been added to enhance the performance of the methods.
2. The authors declared they proposed a framework of face image synthesis by generating methods. Can authors clarify if the framework is suitable for other image generation tech, besides DDPM.
3. The paper is well-structured. However, there are certain grammar and typo errors.
4.  For introduction and related work sections, the description of background can go more specific to synthetic data-based face recognition rather than face recognition.
5. More recent face image synthetic methods should be investigated.
6. The authors have implied that synthetic images can solve privacy and legal issues. Please clarify how your methods can avoid privacy and legal issues, since synthetic images are generated on real-world images.
7. Extra experiments should be conducted to support the methods raised by this work. For example, different settings of fixed t_{0} should be examined, also "baseline + 2-stage-fixed + attn" should be conducted in the ablation experiment.

**Questions:**

1. In this abstract section, can authors explain how the methods raised in this paper preserve the privacy and legality. If this is not related to the research purpose, it is better to exclude it in case of misleading readers.
2. There were more methods proposed upon IDiff and had better performance, for example, “SDFR: Synthetic Data for Face Recognition Competition”.
3. Line 308, ... by leveraging... Carefully check the grammar and typo errors.
4. It is not very clear about timestep interval T analysis in Figure. The interval upperbound of the x-axis is incrementally increased or resets as well the interval lowerbound.
5. For experimentation, different settings of fixed t_{0} should be examined, also "baseline + 2-stage-fixed + attn" should be conducted.

---

> ### Author Response · Authors · 2024-11-21
> **Response to Reviewer JbLs [1/2]**
>
> We appreciate Reviewer JbLs's acknowledgment of the solidity, originality, clarity and significance of our work. We are glad to address your concerns and provide further details.
>
> **W1**:  The motivation behind our approach lies in the diffusion model's different focus of identity-relevant and irrelevant attributes during sampling, which, to the best of our knowledge, has not been revealed or discussed in previous works.
>
> Moreover, our method effectively addresses the diversity gap observed in synthetic versus real face datasets, a gap that has previously led to diminished FR performance. By generates more diverse synthetic datasets, our method achieves state-of-the-art performance with even half the number of identities. Furthermore, when further increasing the number of identities, our method can even match the performance of real datasets. These results demonstrate the scalability and effectiveness of our approach.
>
> **W2**: Since our motivation lies in different denosing preference in different stages of diffusion reverse process, our method can be applied to various diffusion-based generation methods.
>
> Here we choose FPNDM [1] as a representative and conduct more experiments below (the results related to DDIM have already been included in the paper). As demonstrated by the results, UIFace achieves consistent improvements on both DDIM and FPNDM, showing the effectiveness of our method.
>
> | **methods**         | **LFW** | **CFP-FP** | **CPLFW** | **AGEDB** | **CALFW** | **Average** |
> |---------------------|---------|------------|-----------|-----------|-----------|-------------|
> | baseline_with_DDIM  | 98.98   | 92.57      | 87.0      | 88.42     | 90.7      | 91.53       |
> | UIFace_with_DDIM    | 99.27   | 94.29      | 89.58     | 90.95     | 92.25     | 93.27       |
> | baseline_with_FPNDM | 99.18   | 93.03      | 87.7      | 88.92     | 91.18     | 92.00       |
> | UIFace_with_FPNDM   | 99.2    | 94.66      | 89.75     | 91.5      | 92.86     | 93.59       |
>
> ```
> [1] Liu L, Ren Y, Lin Z, et al. Pseudo Numerical Methods for Diffusion Models on Manifolds[C]//International Conference on Learning Representations.
> ```
> **W3**: We will carefully review and correct grammar and typo errors in the later version.
>
> **W4**: We have discussed synthetic-based face recognition in the related work section 2.3 and line 53-91 in the introduction. As you suggested, we will add more discussion of synthetic-based face recognition methods in the revised version.
>
> **W5**: We have compared the latest ECCV24 work Arc2Face in the paper at the time of our paper submission. Here, we add more comparisons of our method with:
> - methods from SDFR: Synthetic Data for Face Recognition Competition [1]
> - latest works from NeurIPS24 including [2] and [3], which are not released at the time of our paper submission.
>
> Our method still outperforms them with fewer images used, demonstrating the effectiveness of our method.
>
> | methods           | from     | img_num | LFW   | CFP-FP | CPLFW | AGEDB | CALFW | Average |
> |-------------------|----------|---------|-------|--------|-------|-------|-------|---------|
> | APhi              | SDFR     | 0.5M    | 97.45 | 80.04  | 78.03 | 84.75 | 89.95 | 86.04   |
> | BOVIFOCR-UFPR     | SDFR     | 0.5M    | 97.53 | 84.37  | 80.07 | 83.90 | 89.38 | 87.05   |
> | ID3               | NIPS24   | 0.5M    | 97.68 | 86.84  | 82.77 | 91.00 | 90.37 | 89.80   |
> | CemiFace          | NIPS24   | 0.5M    | 99.03 | 91.06  | 87.65 | **91.33** | **92.42** | 92.30   |
> | Ours              |      |  0.5M       | **99.27** | **94.29**  | **89.58** | 90.95 | 92.25 |**93.27** |
> |-------------------|----------|---------|-------|--------|-------|-------|-------|---------|
> | IGD-IDiff-Face    | SDFR     | 1M      | 98.07 | 84.76  | 81.28 | 87.60 | 90.60 | 88.46   |
> | CemiFace          | NIPS24   | 1M      | 99.18 | 92.75  | 88.42 | 91.97 | 93.01 | 93.07   |
> | Ours              |        |   1M      | **99.22** | **95.03**  | **90.42** | **92.45** | **93.18** | **94.06**   |
>
> ```
> [1] Shahreza H O, Ecabert C, George A, et al. Sdfr: Synthetic data for face recognition competition[C]//2024 IEEE 18th International Conference on Automatic Face and Gesture Recognition (FG). IEEE, 2024: 1-9.
> [2] Xu J, Li S, Wu J, et al. ID^3: Identity-Preserving-yet-Diversified Diffusion Models for Synthetic Face Recognition[C]//The Thirty-eighth Annual Conference on Neural Information Processing Systems.
> [3] Sun Z, Song S, Patras I, et al. CemiFace: Center-based Semi-hard Synthetic Face Generation for Face Recognition[C]//The Thirty-eighth Annual Conference on Neural Information Processing Systems.
> ```

---

> ### Author Response · Authors · 2024-11-21
> **Response to Reviewer JbLs [2/2]**
>
> **W6**: about privacy and legal issues
>
> As stated in section 3.5, we utilize an additional unconditional face generation model to generate non-existent identities. Based on these identities, we use our method to generate synthetic datasets to train FR models. In other words, all images used to train the final FR model are derived from non-existent identities.
>
> As mentioned in [1, 2], using such synthetic face datasets to train FR models can address privacy and legal issues caused by real datasets, which are directly collected from the internet without the consent of the individuals. And such argument has been widely acknowledged and adopted in rencent works such as [3, 4, 5].
>
> ```
> [1] Regulation P. Regulation (EU) 2016/679 of the European Parliament and of the Council[J]. Regulation (eu), 2016, 679: 2016.
> [2] DeAndres-Tame I, Tolosana R, Melzi P, et al. Frcsyn challenge at cvpr 2024: Face recognition challenge in the era of synthetic data[C]//Proceedings of the IEEE/CVF Conference on Computer Vision and Pattern Recognition. 2024: 3173-3183.
> [3] Kim M, Liu F, Jain A, et al. Dcface: Synthetic face generation with dual condition diffusion model[C]//Proceedings of the ieee/cvf conference on computer vision and pattern recognition. 2023: 12715-12725.
> [4] Qiu H, Yu B, Gong D, et al. Synface: Face recognition with synthetic data[C]//Proceedings of the IEEE/CVF International Conference on Computer Vision. 2021: 10880-10890.
> [5] Boutros F, Grebe J H, Kuijper A, et al. Idiff-face: Synthetic-based face recognition through fizzy identity-conditioned diffusion model[C]//Proceedings of the IEEE/CVF International Conference on Computer Vision. 2023: 19650-19661.
> ```
> **W7**: Here, we conduct more experiments about ablation and different settings of $t_{0}$ as you suggested.
>
> | settings                              | t0   | LFW   | CFP-FP | CPLFW | AGEDB | CALFW | Average |
> |--------------------------------------- |------|-------|--------|-------|-------|-------|---------|
> | baseline + 2-stage-fixed + attn       | 400  | 50    | 50     | 50    | 50    | 50    | 50      |
> | baseline + 2-stage-fixed              | 500  | 98.83 | 92.94  | 88.37 | 89.1  | 91.15 | 92.08   |
> | baseline + 2-stage-fixed + attn       | 500  | 99.15 | 93.84  | 88.48 | 90.3  | 91.78 | 92.71   |
> | baseline + 2-stage-fixed + attn       | 600  | 99.07 | 93.49  | 88.2  | 89.7  | 91.05 | 92.30   |
> | baseline + 2-stage-fixed + attn       | 700  | 98.92 | 92.94  | 88.13 | 88.98 | 90.77 | 91.95   |
> | baseline + 2-stage-adaptive + attn    |      | 99.27 | 94.29  | 89.58 | 90.95 | 92.25 | 93.27   |
>
> It can be observed that for the fixed strategy, as $t_0$ decreases, the accuracy of the final FR model continues to increase. However, when $t_0$ is less than 500, the training of the recognition model collapses (random guess：50%).
>
> This is because a smaller $t_0$ implies a longer first stage in the sampling process (unconditional generation), which enhances diversity but decreases the intra-class consistency (discussed in Line 438-448). The diversity helps improve the recognition performance. However, when $t_0$ is too small, the intra-class consistency of the generated dataset is insufficient, resulting in training collapse.
>
> In contrast, our adaptive strategy outperforms all settings of fixed strategy, without the need to manually select the optimal $t_0$.
>
> **Q4**: Sorry for the unclear presentation of Figure 2. Figure 2 illustrates the intra-class similarity of images generated with different intervals where identity context $c$ is used. For example, [0, 500] on the x-axis represents that, during the entire sampling process where $t$ decreases from 1000 to 0, in the interval of [0, 500] the model is conditioned on identity context $c$, while in the remaining interval [500, 1000] is conditioned on the empty context $c_e$. The trend in the red box indicates that increasing the length of identity context-conditioned interval in the early sampling stages (larger t) has little effect on intra-class similarity, which demonstrates our motivation that in later stage the model recovers the identity-relevant details while restores identity-irrelevant contents in early stage.

---

### Official Review · Reviewer_TRrB · 2024-11-04

**Soundness:** 3
**Presentation:** 3
**Contribution:** 3
**Rating:** 6
**Confidence:** 3

**Summary:**

This paper addresses the challenge of improving synthetic data generation for face recognition (FR) by tackling the issue of context overfitting, which leads to low intra-class diversity and diminished performance. The authors introduce a novel framework named UIFace, designed to enhance intra-class diversity while maintaining identity preservation. The approach begins by training a diffusion model capable of denoising under identity-conditioned contexts or a learnable empty context. The identity context generates images that maintain identity but lack variation, while the empty context leverages the model's intrinsic ability to produce diverse images, albeit with random identities.

**Strengths:**

1. UIFace seems to be closer to the dataset diversity of real datasets than the previous method IDiff-face.
2. better results can be achieved using less amount of synthetic data, and with increasing amount of data even approaching the performance of direct datasets

**Weaknesses:**

1. The main contribution of the paper lies in the observation that LDM learns different aspects during both pre-training and post-training. However, only the analysis based on Figure 2 is currently insufficient, and there is a lack of theoretical discussion.

2. The training and testing times for the two-stage LDM should be provided for comparison with the one-stage methods.

3. The font size of the axis labels in Figure 2 is too small, and it appears that the author has not used \citep and \citet correctly.

**Questions:**

The authors are encouraged to include a theoretical analysis explaining how LDM learns different subjects at different training time steps.

---

> ### Author Response · Authors · 2024-11-21
> **Response to Reviewer TRrB [1/2]**
>
> We are glad that Reviewer TRrB appreciates our work in terms of the improvements in synthetic dataset diversity and face recognition performance. We provide more details to help address your concerns.
>
> **W1**: First, it is important to note that Figure 2 illustrates our key observation: the model initially recovers identity-irrelevant contents and later restores identity-relevant details in sampling process. This observation underpins our adoption of the two-stage denoising strategy in diffusion model's sampling process.
>
> And here we substantiate our observation through analysis in the frequency domain. Firstly, we posit that identity-relevant information predominantly resides in higher frequency components; a notion supported by prior research. Specifically, research [1, 2] shows that despite low-frequency components containing most of the image's energy, removing them does not hinder the performance of FR models, which continue to perform well with only high-frequency components present. These results indicate that the low-frequency components mainly consist of identity-irrelevant contents, thus not affecting the performance of FR models. On the other hand, the high-frequency components contain id-relevant details, facilitating accurate recognition performance with only high-frequency features available.
>
> Secondly, we demonstrate that high-frequency information is restored in the later stage of reverse process, whereas the low-frequency components are restored in relatively early stage. Consider a continuous diffusion process
> $$ \mathbf{x} _t = \mathbf{x} _0 + \int^{t} _{0} g(s) d\mathbf{w} _t .$$
> where $g(s)$ and $d\mathbf{w}_t$ stand for noise schedule and white noise. In frequency domain, we have
> $$ \hat{\mathbf{x}} _t(\omega) = \hat{\mathbf{x}} _0(\omega) + \hat{\mathbf{\epsilon}} _t(\omega) $$
> where $\hat{\mathbf{\epsilon}}_t $ is the Fourier transform of noise term that the expectation $\mathbb{E}[\hat{\mathbf{\epsilon}}_t(\omega)] = 0$ and variance $\mathbb{E}[|\hat{\mathbf{\epsilon}} _t(\omega)|^{2}] = \int^{t} _{0} |g(s)|^2 d\mathbf{w} _t$.
>
> The power spectral density (PSD) of signal $\mathbf{x}_t$ is as follows,
> $$  S _{\mathbf{x}_t}(\omega) = \mathbb{E}[|\hat{\mathbf{x}} _t(\omega)|^{2}] = \mathbb{E}[|\hat{\mathbf{x}} _0(\omega) + \hat{\mathbf{\epsilon}} _t(\omega)|^{2}] $$
> $$ = \mathbb{E}[|\hat{\mathbf{x}} _0(\omega)|^2] + \mathbb{E}[|\hat{\mathbf{\epsilon}} _t(\omega)|^2] + 2*\mathbb{E}[\hat{\mathbf{x}} _0(\omega)]*\mathbb{E}[\hat{\mathbf{\epsilon}} _t(\omega)] $$
> $$ = \mathbb{E}[|\hat{\mathbf{x}} _0(\omega)|^2] + \int^{t} _{0} |g(s)|^2 d\mathbf{w} _t.$$
> The Signal-to-Noise Ratio (SNR) can be written as
> $$ SNR _{\mathbf{x} _t}(\omega) = \frac{|\hat{\mathbf{x}} _0(\omega)|^{2}}{\mathbb{E}[|\hat{\mathbf{\epsilon}} _t(\omega)|^2]} = \frac{|\hat{\mathbf{x}} _0(\omega)|^2}{\int^{t} _{0} |g(s)|^2 d\mathbf{w} _t}.$$
> Equation above indicates that SNR is only dependent on the PSD of the original image $\hat{\mathbf{x}}_0(\omega)$ since the PSD of white noise is constant at different frequency. As a natural signal, PSD of the image is predominantly distributed in the low frequencies. Therefore, we can assume that $$ |\hat{\mathbf{x}}_0(\omega)|^{2} \propto |\omega|^{-\alpha}$$
> So we have $$ SNR _{\mathbf{x} _t}(\omega) \propto \frac{|\omega|^{-\alpha}}{\int^{t} _{0} |g(s)|^2 d\mathbf{w} _t} $$
> The equation above demonstrates that
> - when timestep $t$ is fixed, higher frequency components have a smaller $SNR$
> - as $t$ increases, $\int^{t}_{0} |g(s)|^2 d\mathbf{w}_t$ grows, leading to a decrease in $SNR$ for all frequencies
>
> The process of adding noise to disrupt image can be considered as a process of reducing $SNR$. Therefore, the higher frequency components are obscured for smaller $t$ in the diffusion forward process, hence restored in the later stage of diffusion reverse process, while the low-frequency components are restored in relatively early stage. It is worth noting that, in addition to the analysis above, [3] also provide experimental evidence to support the aforementioned points. Please refer to it for further details.
>
> Combining the above two statements, we can draw a conclusion that the recovery of identity-relevant properties is performed in the later stage of sampling, while identity-irrelevant contents are restored in relatively early stage.
> ```
> [1] Mi Y, Huang Y, Ji J, et al. Duetface: Collaborative privacy-preserving face recognition via channel splitting in the frequency domain[C]//Proceedings of the 30th ACM International Conference on Multimedia. 2022: 6755-6764.
> [2] Mi Y, Huang Y, Ji J, et al. Privacy-preserving face recognition using random frequency components[C]//Proceedings of the IEEE/CVF International Conference on Computer Vision. 2023: 19673-19684.
> [3] Qian Y, Cai Q, Pan Y, et al. Boosting Diffusion Models with Moving Average Sampling in Frequency Domain[C]//Proceedings of the IEEE/CVF Conference on Computer Vision and Pattern Recognition. 2024: 8911-8920.
> ```

---

> > ### Author Response · Authors · 2024-11-21
> > **Response to Reviewer TRrB [2/2]**
> >
> > **W2**: Thanks for suggesting experiments about training and testing times. We would like to clarify a few points about our method and their implications on computational costs.
> >
> > First and foremost, our training process largely adheres to the established protocols of previous DDPM-based methods, such as IDiff-Face [1]. The primary distinction in our training approach is the introduction of an additional empty context $c_e$, which probabilistically replaces the identity context $c$ in training iterations, and thus carries enhanced diversity (Line 256-261). This modification does not introduce the "two-stage strategy" during training phase, and therefore, does not incur any additional training costs.
> >
> > Secondly, during the sampling phase, our two-stage sampling incurs additional costs in two aspects, computing the differences in cross-attention maps between adjacent timesteps (equation 5) in the first stage, and executing Attention Injection (section 3.4) in the second stage, which are minor computational overhead compared to the forward passes of sampling process.
> >
> > Experimental results indicate that the runtime of our method on V100*8 for sampling 0.5 million images increases by ~10% compared to one-stage method idiff-Face [1] (approximately 8 hours for idiff-Face versus 9 hours for our method). This modest increase in computational overhead is justified by the significant performance improvements achieved by our method.
> > ```
> > [1] Boutros F, Grebe J H, Kuijper A, et al. Idiff-face: Synthetic-based face recognition through fizzy identity-conditioned diffusion model[C]//Proceedings of the IEEE/CVF International Conference on Computer Vision. 2023: 19650-19661.
> > ```
> > **W3**: Thanks for pointing out these issues. We will correct them in revised version.

---

### Author Response · Authors · 2024-11-22
**General Response**

We are very grateful for the time and valuable suggestions provided by all reviewers and we have provided an individual response to each reviewer. Here we want to highlight some general issues.

**Contributions**: The contributions of our paper compared to previous works are as follows:
- **motivation**: Our method is motivated by the diffusion model's different focus of identity-relevant and irrelevant attributes of different timesteps during sampling, which, to the best of our knowledge, has not been revealed or discussed in previous works.
-  **techniques**: We propose a novel two-stage sampling strategy and an attention injection method, which can effectively address the diversity gap between synthetic and real datasets by fully leveraging the model’s inherent capabilities. Such a gap caused by context overfitting has significantly hampered the accuracy of previous synthetic-based face recognition methods compared to FR models trained on real datasets.
-  **Experimental contributions**: We quantitatively validate that our method can enhance the diversity of synthetic face datasets. Moreover, by generating more diverse datasets, our method can achieve state-of-the-art performance with even half the number of identities compared to previous methods. When further increasing the number of identities, our method can even match the performance of FR models trained on real datasets, demonstrating the scalability and effectiveness of our method.



**Rebuttal Revision**: Based on the suggestions of reviewers, we have made the following modifications in our rebuttal revision (marked in blue):

- Increase the font size of Figure 2 to make the presentation clearer (Reviewer TRrB)
- Revise all citations to correctly use \citet and \citep (Reviewer TRrB)
- Correct spelling and grammar errors as much as possible (Reviewer JbLs)
- Emphasize the explanation about generating non-existent identities to prevent misunderstandings about privacy issue (Reviewer JbLs)
- Emphasize how our method ensures inter-class discrepancy (Reviewer zDSW)

**More Analysis and Experiments**: We greatly appreciate the reviewers' valuable suggestions to improve the analysis and experiments of our paper including

- More theoretical discussion about our motivation (Reviewer TRrB)
- Analysis about training and testing times (Reviewer TRrB)
- Experiments on more generative methods, comparison with other recent methods and more ablation studies (Reviewer JbLs)
- Involving experiments with a broader range of identities to validate the scalability of our method (Reviewer zDSW)

We have completed the analysis and experiments above and individually responded to each reviewer. These suggestions really enhance the quality of our paper, and we will incorporate them into the main body or appendix in later version.

---

### Meta-Review · Area_Chair_xcN5 · 2024-12-20

**Metareview:**

This paper introduces a technique aimed at improving the diversity of synthetic datasets used for training face recognition (FR) models by presenting an innovative framework that employs a two-stage denoising process. This approach utilizes both identity-specific and non-identity-specific contexts to generate synthetic images, effectively addressing the common issue of context overfitting found in existing synthetic data methods. As a result, the method enhances intra-class diversity and boosts performance. Reviewers appreciate the motivation behind the paper but raise concerns regarding the clarity of the method’s description. The paper suggests that the training pipeline uses a two-stage strategy during the sampling phase but fails to provide a clear explanation of how the losses from the unconditional and conditional branches are calculated and integrated into each stage. After reading the paper and discussion, AC is also confused about the two-stage sampling process. Does each update need Eq(5) to decide (t0) which stage it is? To enhance the paper’s comprehensibility and reproducibility, a more detailed description of this process is essential in the final revision.

**Additional Comments On Reviewer Discussion:**

During the rebuttal period, reviewers highlighted several crucial aspects that required further elaboration, including a more detailed theoretical analysis of the diffusion model’s learning dynamics across different stages, enhanced clarity on the methodology especially concerning the operation of the two-stage framework, the scalability of the model, and considerations related to inter-class discrepancies. They also sought greater detail on the overall training pipeline. In response, the authors made specific adjustments and provided clarifications to tackle these issues. Despite their efforts, the decision was primarily influenced by unresolved ambiguities in the practical implementation details of the model. While the authors endeavored to address the reviewers’ concerns about experiments, the fundamental issues affecting the paper’s clarity and reproducibility remained unaddressed. Please put a huge effort into adopting reviewers' suggestions for the camera-ready version.

---

### Decision · Program_Chairs · 2025-01-22

Accept (Poster)